# Intensification of the SFE Using Ethanol as a Cosolvent and Integration of the SFE Process with sc-CO_2_ Followed by PLE Using Pressurized Ethanol of Black Soldier Fly (*Hermetia illucens* L.) Larvae Meal—Extract Yields and Characterization

**DOI:** 10.3390/foods13111620

**Published:** 2024-05-23

**Authors:** Vanessa Aparecida Cruz, Nilson José Ferreira, Elise Le Roux, Emilie Destandau, Alessandra Lopes de Oliveira

**Affiliations:** 1High-Pressure Technology and Natural Products Laboratory (LTAPPN), Department of Food Engineering, Faculty of Animal Science and Food Engineering, University of São Paulo, 225 Duque de Caxias Norte Avenue, Pirassununga 13635-900, SP, Brazil; vanessa_ap_cruz@usp.br (V.A.C.); nilsonferreira@usp.br (N.J.F.); 2Institut de Chimie Organique et Analytique, Université d’Orléans, CNRS, UMR 7311, BP6759, Orléans Cedex 2, 45067 Orléans, France; elise.le-roux@univ-orleans.fr (E.L.R.); emilie.destandau@univ-orleans.fr (E.D.)

**Keywords:** sustainable extraction, extraction processes, insect, fatty acids, bioactive compounds, anthropoentomophagy

## Abstract

The objective of this research was to investigate and compare the results obtained in the intensification and integration of (sc-CO_2_) under different pressure conditions (25 and 30 MPa) at 60 °C. When intensifying the process, ethanol (10%) was used as a co-solvent (sc-CO_2_ + EtOH). In the process integration, black soldier fly larvae flour, defatted via supercritical extraction (SFE), was the raw material for pressurized liquid extraction (PLE) using ethanol as solvent. The extract yields, fatty acid profile, free fatty acids, triacylglycerols (TAGs), oxidative stability, and nutritional quality of the oil obtained using sc-CO_2_ + EtOH were evaluated. The composition of bioactive compounds (carotenoids, acidity, antioxidant compounds, tocopherols, and phospholipids) was determined in both extracts. The yields of the extracts were different by 32.5 to 53.9%. In the extracts obtained with sc-CO_2_ + EtOH (10%), the predominant fatty acids were oleic, palmitic, and linoleic, with considerable levels of desirable fatty acids (DFA), tocopherols, and phospholipids. The nutritional indices showed good values for polyunsaturated and saturated fatty acids (PUFAs/SFAs), above 0.45%. Extracts from larvae meal defatted with SFE showed carotenoids, phenolic compounds, and antioxidant activity. HPTLC and HPLC analyses indicated the presence of amino acids, sugars, phenolics, and organic acids in their composition. This study revealed that the supercritical fluid extraction (SFE) process, or its conditions, can modify the fatty acid composition and the presence of minor bioactive compounds in the obtained extracts.

## 1. Introduction

In the future, there will not be enough protein food for the entire population, which in 2050, will be around 10 billion people [1]. Therefore, insects are gaining more and more space in animal (entomophagy) and human (anthropoentomophagy) food, especially in countries in which anthropoentomophagy is not traditional. Recognized for their nutritional benefits, insects represent a sustainable alternative to traditional animal proteins. Efforts have been made to increase the consumption of insects and their products, mainly by the Food and Agriculture Organization of the United Nations (FAO), which has greatly encouraged their consumption due to their nutritional and functional values and the benefits that protein production provides via insects to the environment because they do not require large cultivation areas or water resources [1]. Another advantage of raising insects is the very low emission of polluting gases and the fact that they can be fed with food waste from restaurants, making insects a highly sustainable and environmentally beneficial food [2].

The European Union has approved for human consumption some insects for consumption in full or via derivative products, and other insects, such as *Gryllodes sigillatus*, *Alphitobius diaperinus*, and *Hermetia illucens* L., are in progress, awaiting approval by the European scientific committee [3].

Black soldier fly larvae meals (*Hermetia illucens* L.) have the potential to be included in human nutrition due to their high levels of proteins, amino acids, fibers, and lipids. The larvae meal of these insects can present lipid contents of up to 40% dry weight, being considered the second largest macronutrient in its composition [4]. The oil of black soldier fly larvae, obtained via supercritical fluid extraction, shows varying levels of monounsaturated (28.63%) and polyunsaturated fatty acids (22.94%), in addition to saturated fatty acids (48.43%) [5].

Insects in the pupal and larval stages have higher concentrations of lipids, which is why they are important sources of energy, in addition to having essential fatty acids, such as linoleic (ω-6) and linolenic (ω-3), in their composition; due to these characteristics, they can be used to combat child malnutrition [6]. However, there is still a limited number of studies and applications involving insect fats in human nutrition.

A variety of insect oil extraction techniques have been studied, with the majority of research focusing on the use of supercritical fluid extraction (SFE) with carbon dioxide (CO_2_) to obtain edible insect oils. Supercritical carbon dioxide (sc-CO_2_) offers several advantages as a solvent, including low toxicity and high solubilization power, resulting in high mass transfer rates. It also requires low processing temperatures due to its low critical temperature (31 °C). Consequently, it does not degrade bioactive compounds, as is often observed in conventional extractions [7]. In addition, it eliminates the need for multiple sample preparation procedures for extraction, speeding up the process. Furthermore, CO_2_ is cost-effective and can be recovered and recycled in the process, making it an environmentally sustainable option [8,9].

To alter the polarity of sc-CO_2_ in the extraction process to obtain extracts rich in polar compounds, cosolvents can be used. For instance, ethanol can be used in the extraction of oils via sc-CO_2_ when the desired product contains an abundance of minor compounds, such as tocopherols, phenolic compounds, and pigments, among others. This approach helps ensure the use of solvents that are generally recognized as safe (GRAS) in the process [7].

Another unconventional technique that has been used in oil extractions is pressurized liquid extraction (PLE) using ethanol as solvent. The efficient extraction of oil using ethanol is only possible under high pressures. Ethanol is the most commonly used solvent in process studies because it is GRAS and sustainable [10,11]. The fact that it is a polar solvent implies that the obtained oil will be rich in phospholipids and other minor compounds [12,13].

PLE is a solid-liquid extraction process that is frequently used in intermittent processes. This type of extraction normally takes place at temperatures above the solvent’s boiling point and under high pressures (≈10 MPa), which enhances the efficiency of the extraction process by forcing the entry of the solvent into the matrix [14]. High temperatures reduce the viscosity and increase the diffusivity of solvents, which, consequently, result in increased mass transfer rates and enhanced solubility of the compounds present in the matrix. These process characteristics, when compared to traditional methods, result in shorter extraction times and reduced solvent usage in intermittent processes.

When used in an intermittent process, PLE does not require high concentrations of solvent and can be performed in less time, compared to conventional methods [15].

Both SFE and PLE are innovative methods for obtaining vegetable and animal oils, being considered green and sustainable processes. SFE, in particular, is regarded as a clean technology because it does not leave any organic solvent residues in the extracts. This is because, under ambient or low-pressure conditions, CO_2_ is a gas and releases the extracts in pure form.

In this context, the aim of this study was to assess the composition of black soldier fly larvae oil obtained under different extraction conditions using sc-CO_2_, both with and without the incorporation of ethanol as a cosolvent. Insect meal, defatted via SFE, served as the raw material for PLE to obtain an ethanolic extract for composition analysis. Following the extractions, the protein concentrate from the defatted black soldier fly larvae meal was prepared for use in food formulation.

The integration of these processes aligns with the concept of microbiorefinery, a bioactive production system that utilizes green and sustainable solvents that, by way of technology intensification or integration, generate clean products (extracts and co-products free of toxic organic solvent residues) that can be potentially applied in the production of foods, pharmaceuticals, and cosmetics.

The obtained extracts were evaluated regarding their yields, fatty acid content, nutritional quality indices, oxidative stability, chemical composition, antioxidant activity, and bioactive compounds.

## 2. Materials and Methods

### 2.1. Purification of Black Soldier Fly Larvae Meal

The dried larvae of *Hermetia illucens* Linnaeus were supplied by the company Verde Paisagem e Jardinagem Ltd.a. in Montes Claros, MG, Brazil (latitude, 16°43′41″, longitude, 43°51′54″, and altitude, 638 m). Before being purified, they were degreased using the supercritical CO_2_ extraction process reported by Cruz et al. [5].

Black soldier fly larvae flour presented a proximate composition on a dry basis as follows: moisture 3.60%, mineral matter 18.47%, crude protein 30.66%, crude fiber 8.42%, ethereal extract 32.70%, and non-nitrogen extract 6.18%, as reported by Cruz et al. [5].

The purification of the black soldier fly larvae meal was conducted with the aim of generating two distinct extracts, one rich in oil and the other rich in polar compounds. The integration of processes that use green solvents enabled obtaining three different products, including insect meal oil, an extract rich in polar compounds obtained via pressurized ethanol, and a protein concentrate powder for dietary supplementation (Figure 1).

#### 2.1.1. Extraction with Supercritical CO_2_ (sc-CO_2_) Using Ethanol as a Cosolvent

The extraction experiments using sc-CO_2_, with and without ethanol as a cosolvent, were carried out under two pressure conditions (25 and 30 MPa) at a temperature of 60 °C, with a constant flow rate of 10 g/min for 90 min, respecting a static time of 20 min according to the process optimization proposed by [5]. The equipment used was an SFC/RESS (Thar Instruments Co./Waters, Pittsburgh, PA, USA), and the extracts were collected in 100 mL glass vials immersed in an ice bath.

The ethanol flow rate remained constant (1.13 g/min) throughout the extraction, maintaining a 10% cosolvent ratio. In the 290 cm^3^ extractor, 10 g of dried and crushed black soldier fly larvae meal was packed between inert glass spheres measuring 5 mm in diameter to completely fill the extraction cell. At the end of the extraction process, the ethanol was evaporated in a rotary evaporator (Marconi, MA-120, Piracicaba, SP, Brazil) at 50 °C for 20 min.

The oil and defatted meal were weighed and stored at −20 °C for further analyses. All extractions were performed in triplicate. The overall yield (*X*_0_) was determined using the ratio between the mass of the extracted matter (*m*_0_) and the mass of the fed matter on a dry basis (*fm_db_*), calculated using Equation (1), as follows:(1)$$X_{0}=\frac{m_{0}}{{fm}_{db}}\times 100$$

#### 2.1.2. Obtaining Defatted Black Soldier Fly Larvae Meal Extract via PLE in an Intermittent Process Using Ethanol as Solvent

The black soldier fly larvae meal, defatted in the extraction process with sc-CO_2_, without cosolvent, was subjected to pressurized liquid extraction (PLE) using ethanol as solvent. The extraction was carried out in a Dionex ASE 150 system (Thermo Fisher Scientific, Newington, CT, USA).

The defatted meal (7 g) was placed in a fixed bed extractor (34 cm^3^) that had been previously sealed at one end (outlet) and contained a paper filter at its base. The pressurized solvent used was ethanol (99.5%), and the extraction conditions optimized for phenolic compounds, adopted from Rodrigues et al. [15], were a temperature of 100 °C, a rinse volume (RV) of 120% of the extraction cell volume of 34 mL, equivalent to 40.8 mL, and 4 cycles (C) with a static time (St) of 6 min. Static time is defined as the duration of contact between the solvent and the matrix in each cycle (C) before the purging of the extract at the end of each cycle for the concomitant entry of the new rinse volume (RV).

In the PLE in an intermittent process, the extraction cell was packed with the defatted meal and placed in an oven with a controlled temperature. Ethanol was then pumped into the cell to fill any empty spaces until, due to the action of temperature and the pumped solvent, the system reached a pressure of 10.35 MPa. Once the equilibrium between T and P was established in the extraction cell, the first cycle (C) began, and after a St of 6 min, the extract was purged for the first time. During the purge, the extract was collected, and the pump simultaneously sent ethanol to the extractor, maintaining constant P and T conditions. Considering a rinse volume of 40.8 mL, the pump sent 10.2 mL of solvent into the cell in each cycle. After the purge and with the introduction of solvent, the static time count restarted until the next cycle was completed. At the end of the last cycle, a final purge with N2 (100 s) was conducted to collect the remaining extract in the extraction cell.

The solvent of the ethanolic extract of the defatted black soldier fly larvae meal was evaporated in a rotary evaporator (Marconi, MA-120, Piracicaba, SP, Brazil) at 50 °C for 40 min. The extract and the purified meal were then weighed and stored at −20 °C for further analyses. All extractions were performed in triplicate.

### 2.2. Characterization of the Extracts

Some of the analyses were carried out with only the oil obtained via SFE using sc-CO_2_ and ethanol as a polarity modifier, while others were performed with both the oil and the ethanolic extract of the defatted meal obtained via PLE, mainly in the analysis that quantified minor compounds and antioxidant activity.

#### 2.2.1. Fatty Acid Profile and Probable Triacylglycerols (TAGs) in the Oil Obtained via Extraction with sc-CO_2_ Using Ethanol as a Cosolvent

The fatty acid profile was determined using an adaptation of method 972.28 [16]. A total of 100 mg of oil extracted from black soldier fly larvae meal was weighed and placed in a Falcon tube, to which 4.0 mL of 0.5 N NaOH solution in methanol was added. This solution was heated to 100 °C and shaken for 5 min. Next, 5.0 mL of the esterification reagent methanolic boron trifluoride (BF3) was added, and the tube was again heated and shaken at 100 °C for 2 min. Afterward, 5.0 mL of chromatographic-grade hexane was added, followed by another round of heating at 100 °C for 1 min. Subsequently, 15 mL of saturated NaCl solution (35.7 g NaCl/100 mL H_2_O) was added, and the tube was shaken for 30 s in a tube shaker. After that, the solution was maintained at rest until complete phase separation. Finally, the supernatant was collected, and Na_2_SO_4_ was added to remove moisture, followed by drying with N_2_ in a sample concentrator. The methyl esters were diluted in hexane to 2.0 mL. The hexane containing the fatty acid methyl esters was carefully transferred into injection vials.

This solution of fatty acid methyl esters in hexane was analyzed using gas chromatography coupled with mass spectrometry (GC/MS). A QP2010 Plus GCMS device (Shimadzu, Tokyo, Japan) equipped with an automatic injector (AOC-5000, Shimadzu, Tokyo, Japan) was used in the analysis. A capillary column with a biscyanopropylsiloxane stationary phase (100 m × 0.25 mm id × 0.20 μm df, SP-2560 Supelco, Bellefonte, PA, USA) was also used. Helium was used as the carrier gas at an internal velocity of 28.8 cm/s. Injection was performed in split mode (rate of 12.5), with an injected volume of 1.0 µL. Both the injector and detector were set at 250 °C. The initial temperature of the oven was 100 °C for 1 min, which was then raised to 195 °C at a rate of 5 °C/min and to 250 °C at a rate of 20 °C/min. The fatty acid methyl esters were identified by comparing the mass spectra of the compounds with those stored in the NIST Library integrated into the GC/MS system (NIST 11 and NIST 11s).

The probable triacylglycerol (TAG) profile of the *Hermetia illucens* L. oil was determined based on the fatty acid composition by way of a statistical algorithm using the MATLAB R2013A Inc. (Natick, MA, USA) computational prediction software, as suggested by [17].

#### 2.2.2. Identification of Free Fatty Acids in the Oil Obtained via Extraction with sc-CO_2_ Using Ethanol as a Cosolvent

The determination of free fatty acids present in the oil obtained from black soldier fly larvae meal was conducted according to the method described by [18], with some modifications. This technique assumes that to determine the free fatty acid (FFA) content, it is necessary to ensure the complete neutralization of all FFAs present in the oil.

A total of 3.0 g of oil was weighed, to which 5.0 mL of NaOH solution (12.69%, m/m) was added. The NaOH + oil mixture resulted in the formation of fatty acid salts. It was then centrifuged to separate the soap. Following centrifugation, three phases were observed. The aqueous and oily phases were drained, while the solid phase (soap) was carefully transferred into a Falcon tube. The soap was then mixed with hexane (5.0 mL), shaken, and centrifuged again. This process was repeated 3 times, carefully collecting the upper phase (hexane), in which the salts are found, which were dried under N_2_ flow.

After obtaining the fatty acid salts, esterification was performed for compound volatilization. A total of 3.0 mL of methanolic boron trifluoride (BF3) was added, and the mixture was heated to 100 °C for 2 min, followed by immediate cooling under running water. Next, 6.0 mL of hexane (chromatographic grade) was added, followed by the addition of Na_2_SO_4_ in the supernatant phase mixture (hexane + fatty acid esters) to remove moisture. The mixture was then analyzed using gas chromatography coupled with mass spectrometry, as described in the previous item.

#### 2.2.3. Nutritional Quality Indices of the Lipids

The nutritional value indices of the lipids extracted from *Hermetia illucens* L. were defined based on their fatty acid composition, as proposed by [19]. The ratio between polyunsaturated and saturated fatty acids (PUFAs/SFAs) was determined, as well as the thrombogenic index (TI), which estimates the availability for the development of clots in blood vessels (Equation (2)); the atherogenic index (AI), which measures the tendency for the formation of atheromas in the internal walls of arteries (Equation (3)); the hypocholesterolemic/hypercholesterolemic ratio (HH) or hypocholesterolemic fatty acids (OFA) index, which measures the increase in blood cholesterol (Equation (4)); the hypocholesterolemic index (DFAs = h), which indicates the probability of reducing blood cholesterol (Equation (5)); the nutritional value index (NVI) (Equation (6)); and the proportion of hypocholesterolemic and hypercholesterolemic fatty acids (h/HH) (Equation (7)). This method also enables the determination of oxidizability through essential fatty acid quantification (COX) (Equation (8)) [19].
TI = ((C14:0 + C16:0 + C18:0))/([(0.5 × MUFA) + (0.5 × Σ ω-6) + (3 × Σ ω-3) + (Σ ω-3/ω-6)])(2)
where MUFA is the sum of monounsaturated fatty acids, and ω-6 and ω-3 are polyunsaturated fatty acids.
AI = ([C12:0 + (4 × C14:0) + C16:0])/(Σ UFA)(3)
where UFA is the sum of unsaturated fatty acids.
OFAs = C12:0 + C14:0 + C16:0 (4)
DFA = “UFA” + C18:0(5)
NVI = (C18:0 + C18:1)/C16 (6)
h/HH = ((C18:1ω-9 + C18:2ω-6 + C18:3ω-3))/((C12:0 + C14:0 + C16:0)) (7)
where ω-9, ω-6, and ω-3 are essential fatty acids.
COX = ([C18:1 + (10.3 × C18:2) + (21.6 × C18:3)])/100(8)

When calculating the AI (Equation (3)), the fatty acid myristic acid (C14:0) is considered 4 times more atherogenic than the other fatty acids, which is why it is attributed a coefficient of 4. The fatty acids C18:1, ω-6 (oleic acid), and the remaining monounsaturated acids are assigned coefficients of 0.5 (Equation (2)) because they are less antiatherogenic than ω-3 fatty acids, which are assigned a coefficient of 3 (Equation (2)) [20,21,22].

#### 2.2.4. Oxidative Stability of the Oil Obtained via Extraction with sc-CO_2_ Using Ethanol as a Cosolvent

Oxidative stability was determined according to [23] method Cd 12b-92, using Biodiesel Rancimat 873 equipment (Metrohm, Herisau, Switzerland). In this analysis, approximately 2.5 mL of the oil was transferred into tubes and placed in a heating block at 120 °C, where they were subjected to an airflow of 20 L/h.

#### 2.2.5. Determination of Acidity of the Oil Obtained via SFE with sc-CO_2_ and Ethanol as a Cosolvent and of the Ethanolic Extract Obtained via PLE

The total titratable acidity (TA) was determined for all samples according to the Ca 5a-40 method [23]. Sample acidity corresponded to the levels of free fatty acids (FFAs) present, mainly in the oil obtained via sc-CO_2_ using ethanol as solvent, but also in the ethanolic extract obtained after the insect meal was defatted via SFE. The FFA content was calculated according to Equation (9), as follows:% FFA = ((V × M × 56.1))/m(9)
where % FFA represents the percentage of free fatty acids, V is the volume of solution used in titration, M is the molarity of the KOH solution, the molar mass of KOH is 56.1, and m represents the mass of the crude extract sample. The results were expressed in KOH content (%, m/m), and all analyses were performed in triplicate.

#### 2.2.6. Total Phenolic Content (TPC) of the Oil Obtained via Extraction with sc-CO_2_ Using Ethanol as a Cosolvent and the Ethanolic Extract Obtained via PLE

The TPC of the oil and the ethanolic extract of black soldier fly larvae meal was quantified according to the methodology described by [24]. After mixing the samples with the reagents (Folin-Ciocalteu reagent and sodium carbonate) and allowing them to react for 2 h at ambient temperature (24 ± 0.5 °C), the absorbance was read at 760 nm using a UV-Vis spectrophotometer (Thermo Fisher Scientific, Genesys 10 S, Waltham, MA, USA). The TPC was calculated using gallic acid as a standard at concentrations of 0, 10, 20, 30, 40, 50, and 60 mg/L (R^2^ = 0.9999), and the results were expressed in mg of gallic acid equivalent (GAE)/g of oil.

#### 2.2.7. Determination of Carotenoid Content in the Oil Obtained via Extraction with sc-CO_2_ Using Ethanol as a Cosolvent and in the Ethanolic Extract Obtained via PLE

In this analysis, the samples were dissolved in a proportion of 0.5 mL of oil or ethanolic extract of black soldier fly larvae meal in 3.0 mL of petroleum ether. The total carotenoid content is determined based on beta-carotene because the adopted molar absorption coefficient (A1% 1 cm) of 2592 refers to beta-carotene solubilized in petroleum ether [25].

For some samples, the turbidity caused when the oil was mixed with petroleum ether was resolved by adding a small amount of NaCl to remove moisture. The mixture was then centrifuged for 5 min to remove the NaCl, followed by the direct reading of the clear supernatant.

The samples were placed in 5 mL cuvettes and read using a UV-Vis spectrophotometer (Thermo Fisher Scientific, G10S UV-Vis, Shanghai, China) at 450 nm. The total carotenoid content was calculated according to Equation (10), as follows:(10)$$Carotenoid\;content\;(mg/100g)=\frac{A\times V\times 10^{6}}{A_{1cm}^{1\%}\times M\times 100}$$
where *A* is the absorbance of the solution at a wavelength of 450 nm for beta-carotene, *V* represents the final volume of the solution (mL), $A_{1cm}^{1\%}$ is the coefficient of molar absorptivity of the oil or extract in a given solvent (petroleum ether is 2592 for beta-carotene), and *M* represents the mass of the sample taken for analysis.

#### 2.2.8. Tocopherol Content in the Oil Obtained via Extraction with sc-CO_2_ Using Ethanol as a Cosolvent and in the Ethanolic Extract Obtained via PLE

The determination of tocopherol (vitamin E) content was carried out according to method Ce 8–89 [23], in which the oil and ethanolic extract of black soldier fly were weighed (3.0 g) and mixed with hexane in an extract-to-solvent ratio of 1:3, using an OS-100 orbital shaker (HiLab, Bahasa, Indonesia) at 280 rpm for 60 min, followed by maceration for 24 h. Subsequently, the mixture was centrifuged for 5 min at 5000 rpm, and the supernatant was filtered using Whatman N° 1 filter paper and concentrated using a rotary vacuum evaporator (Büchi RE 121, Flawil, Switzerland). A 20 μL aliquot of the concentrated extract was used in the HPLC analysis with a 2475 fluorescence detector (Alliance 2697, Waters, Milford, MA, USA).

#### 2.2.9. Determination of Phospholipid Content in the Oil Obtained via SFE with sc-CO_2_ Using Ethanol as a Cosolvent and in the Ethanolic Extract Obtained via PLE

The phosphorus content was determined using inductively coupled plasma optical emission spectrometry (ICP-OES; Perkin-Elmer, Optima 5300 DV model, Waltham, MA, USA) following the Ca 20–99 official method [23]. The amount of phosphorus was converted into phospholipid equivalents by multiplying the measured phosphorus content by a conversion factor of 30, as determined by the [23] Ca 12–55 official method.

#### 2.2.10. Preliminary Determination of the Classes of Compounds Present in the Ethanolic Extract of Defatted Black Soldier Fly Larvae Meal Obtained via PLE

High-performance thin-layer chromatography (HPTLC) was used for preliminary analyses of the ethanolic extract composition obtained from defatted soldier fly larvae flour using PLE. HPTLC silica gel 60 plates with fluorescent indicator F254 (Supelco, Mississauga, ON, Canada) were used as the stationary phase. Sample deposition was carried out using CAMAG automatic TLC Sampler 4, sample elution was carried out in a CAMAG ADC2 automatic developing chamber, and photography was carried out using a CAMAG TLC visualizer 2. The data were processed using vision CATs CAMAG software visionCATS 4.0 (4.0.24047.1). When plate derivatization was necessary, the CAMAG Derivatizer was used, with specific recommendations for each reagent.

In the analysis of the presence of sugars and amino acids, acetonitrile/water (75:25) was used as the mobile phase. To analyze the presence of phenolic compounds, ethyl acetate/water/formic acid/acetic acid (100:27:11:11) was used as the mobile phase; as well as the mobile phase normally used for the analysis of organic acids, ethanol/ammonium hydroxide/water (75.5:12.5:12), and in the analysis of condensed tannins, the mobile phase was composed of ethyl acetate/methanol/water (79:11:10). After eluting the samples, the developers were prepared [26] and sprayed on the silica plates (CAMAG Derivatizer), specific to each analysis. Ninhydrin (100 mg ninhydrin dissolved in 100 mL ethanol spraying followed by drying at 105 °C/3 min) was used to identify the presence of amino acids and Molish (2 g α-naphtol solubilized in 100 mL ethanol and sulfuric acid at 5% in ethanol spraying and drying at 120 °C/5 min) to identify the presence of sugars. To identify phenolic compounds, Neu (1 g of diphenyl boric acid ethylamino ether in 100 mL of MeOH) developer was applied, followed by PEG (polyethylene glycol (PEG) 4000 at 5%) in EtOH with reading taken at 366 nm. To identify condensed tannins, the developer Vanillin-HCL was used (spraying and drying at 100 °C/10 min).

UHPLC was also used in the preliminary analysis of the family of compounds present in the hydroalcoholic extract. UHPLC analyses were performed with the Thermo Scientific Ultimate 3000 RSLC analytic system and the Chromeleon 7 software. The column used was the Luna Omega C18 (150 mm, 0.1 mm, 1.6 µm) column. The mobile phase used was water acidified with 0.1% formic acid (A) and acetonitrile acidified with 0.1% formic acid (B) with the following gradient: 0–2 min: 5% of B; 2–22 min: 5–100% of B; 22–31 min: 100% of B; 31–31.5 min: 5% of B; and 31.5–41 min: 5% of B. The flow rate was fixed at 0.4 mL/min. Samples were prepared at a concentration of 16 mg/mL in EtOH, and standards were prepared at a concentration of 1 mg/mL in EtOH for benzoic acid and water for gallic acid and pyrocatechol. A total of 2 µL of sample and standards was injected, and the column was heated to 40 °C. Detection was performed with a DAD visualized at 280 mn and an ELSD detector.

### 2.3. Antioxidant Activity of the Oil Obtained via Extraction with sc-CO_2_ Using Ethanol as a Cosolvent and the Ethanolic Extract Obtained via PLE

#### 2.3.1. Antioxidant Activity Measured using the Reduction in the Free Radical DPPH

Antioxidant activity is based on hydrogen donation or free radical capture using radical 2,2-diphenyl-1-picrylhydrazyl (DPPH). DPPH is a free radical that can be obtained directly by dissolving the reagent in an organic medium.

According to the adopted methodology, with minor modifications, a total of 100 μL of oil or the ethanolic extract obtained from the defatted black soldier fly larvae meal was placed in test tubes together with 3.9 mL of DPPH radical solution. The tubes were homogenized in a tube shaker. Afterward, the reaction took place for 80 min in a dark environment at room temperature. The blank used for calibrating the UV-Vis spectrophotometer (Jenway 7305 Spectrophotometer, Stone, Staffordshire, Germany) at an absorbance of 515 nm was ethanol. The readings of the samples and the Trolox calibration curve were carried out at an absorbance of 515 nm. Antioxidant activity was calculated according to Equation (11) and expressed in % of DDPH inhibition [27].
% of DPPH inhibition = ((Abs control − Abs sample) × 100) (Abs control)(11)

#### 2.3.2. Antioxidant Activity Measured by the Reduction in the Free Radical ABTS

In this analysis, the radical 2,2’-azinobis(3-ethylbenzothiazoline)-6-sulfonic acid radical cation (ABTS) was diluted in ethanol until it reached an absorbance of 0.70 (±0.02) at a wavelength of 740 nm at ambient temperature. In one test tube, 30 μL of the oil or ethanolic extract obtained from defatted black soldier fly larvae meal was added, along with 3.0 mL of ABTS solution. After 6 min of incubation at ambient temperature, the samples were read at 740 nm using a UV-Vis spectrophotometer (Jenway 7305 Spectrophotometer, Germany).

In this method, the radical ABTS (ABTS•+) is generated in the reaction between ABTS and potassium persulfate. With the addition of antioxidant compounds, the radical ABTS•+ is reduced to ABTS, promoting the loss of color of the reaction medium.

The antioxidant activity of the samples was expressed in mM TE/100 g of oil or ethanolic extract [28].

### 2.4. Statistical Analyses

All statistical analyses were conducted using analysis of variance (ANOVA), with a significance level of 5%, followed by Tukey’s mean comparison tests (*p* < 0.05). The data were processed using the SAS statistical package, version 8.0 (Statistical Graphics Corp., Cary, NC, USA).

## 3. Results

### 3.1. Yield of the Oil Obtained via SFE with sc-CO_2_ Using Ethanol as a Cosolvent

The chemical composition of the dried and crushed whole black soldier fly larvae meal consisted of 32.7% lipids and 30.7% protein [5].

The yields of black soldier fly larvae meal oils obtained via SFE with sc-CO_2_ and ethanol as a cosolvent were 32.5% and 36.1% at 60 °C and pressures of 25 and 30 MPa, respectively. The yield values differed, indicating that higher pressures result in greater extractions (Table 1). In this case, the density of CO_2_ had a positive influence because, at a constant temperature, higher pressures increase the solvent’s density, thus enhancing its solubilization power.

In the SFE process, the use of ethanol as a cosolvent resulted in higher yields. When only sc-CO^2^ was used in the extraction, the oil yield was lower, with 31.0 and 33.0% at 25 and 30 MPa, respectively [5]. Considering that the oil content in the meal was 32.7%, the SFE with ethanol as a cosolvent generated an extract with a recovery of up to 106.5% (Table 1), indicating that other compounds were also extracted, in addition to nonpolar compounds.

### 3.2. Yield of the Ethanolic Extract of the Defatted Meal Obtained via PLE

The defatted meal obtained via SFE* with only sc-CO_2_ [5], subjected to pressurized ethanol extraction, resulted in an extract yield of 20.9%, (ethanolic extract). The combined yield of the two extracts reached a total yield of 53.9%. This integration of the processes, which involved SFE with sc-CO_2_ followed by PLE with pressurized ethanol (Oil via SFE* + ethanolic extract via PLE), generated two distinct extracts separately. When considering the total recovery of the extract to the lipid levels found in whole black soldier fly larvae meal (32.7%), the recovery of compounds reached 157% (Table 1), i.e., in addition to the extraction of total fat, other compounds were also obtained, which were analyzed.

The defatted black soldier fly larvae meal extract, obtained via PLE, exhibited gel-like characteristics and low lipid content. Because ethanol is a polar solvent, it extracted a wide range of classes of compounds, including polyphenols and phenolic compounds, among others.

The intensification of the oil extraction process from black soldier fly larvae meal with sc-CO_2_ was achieved using ethanol as a cosolvent. In this case, there was a recovery of compounds greater than 100% (106.5%, Table 1) when compared to the oil content in the black soldier fly larvae meal. This indicates that, in addition to lipid compounds, other minor compounds were also extracted along with the oil, primarily due to the change in solvent polarity. The use of polarity modifiers, such as ethanol, with the aim of extracting minor active compounds in lipid extractions using sc-CO_2_ has been extensively studied, even in insect meals [4]. One study evaluating the influence of the extraction of active minor compounds in insect oils using different solvents also demonstrated the efficiency of polar solvents in the extraction of various classes of lipids [6].

Along with the oils of *Acheta dometicus* (house cricket) and *Alphitobius diaperinus* (lesser mealworm beetle), compounds with antioxidant activity, phenolic compounds, and carotenoids were also obtained in extractions using polar solvents [12,29].

Cantero-Bahillo et al. [4], while investigating extracts obtained from *Hermetia illucens* L. larvae meal using sc-CO_2_ and ethanol as a cosolvent, achieved a yield of 43%, which was higher than the yield observed in this study. This difference can be attributed to the high pressure used (45 MPa), the higher ethanol concentration (20%), and the greater CO_2_ flow rate (100 g/min). It is important to highlight that the centesimal composition of insect larvae meals may vary depending on the cultivation and feeding conditions of the larvae.

The oils of yellow mealworm (*Tenebrio molitor*) larvae meal and house cricket (*Acheta domesticus*) larvae meal, extracted via PLE using ethanol as a solvent, also showed higher yields (32.37 and 24.85%) when compared to other organic solvents in the study conducted by Otero et al. [30].

The efficiency in obtaining extracts from black soldier fly larvae meal, *Tenebrio molitor*, and *Acheta domesticus* via PLE using ethanol as solvent exhibited similar results to those found in this study in terms of extraction yield. The pursuit of knowledge regarding the composition of insect meal extracts has grown, as has the exploration of new processes, such as PLE with the use of green and sustainable solvents and the use of pressurized ethanol, which has demonstrated its efficiency [31]. Insect oils rich in bioactive compounds enhance nutritional composition and can be applied in the food industry.

### 3.3. Characterization of the Extracts

#### 3.3.1. Fatty Acid Profile and Probable Triacylglycerol (TAG) Content in the oil Obtained via Extraction with sc-CO_2_ Using Ethanol as a Cosolvent

The fatty acid profile of *Hermetia illucens* L. oil extracted via SFE with sc-CO_2_ and ethanol as a cosolvent showed a very similar profile to that of black soldier fly larvae oil obtained via SFE without cosolvent [5]. The predominant fatty acids in both oils were oleic acid (18:1, O), followed by palmitic (16:00, P) and linoleic acids (18:2, Li). Meanwhile, the fatty acids capric acid (10:0, C), palmitoleic acid (16:1, Pa), stearic acid (18:0, S), and linolenic acid (18:3 ω-3, Ln) were found in lower percentages (Table 2). Solvents can alter the concentrations of fatty acids in oils due to their polarity, mainly because polar solvents extract higher levels of free fatty acids. According to [32], polar solvents, such as ethanol, can extract greater levels of linoleic acid, whereas nonpolar solvents extract higher concentrations of oleic and palmitic acid. In the present study, the SFE of the black soldier fly larvae oil did not present differences in fatty acid composition when ethanol was used as a cosolvent (Table 2).

When analyzing the influence of pressure values as a variable of the SFE process with sc-CO_2_ and ethanol as a polarity modifier, it can be observed that the saturated fatty acid (SFA) contents differed from each other (Table 2), with higher pressure (30 MPa) resulting in a higher concentration of SFAs. In SFE, the higher the pressure, the greater the solubilization power of the solvent; therefore, elevated pressures can make the solvent less selective in the process [33], leading to the extraction of larger amounts of compounds with higher solubility, such as SFAs. This behavior does not apply to mono and polyunsaturated fatty acids (Table 2).

Insect oils have shown similar fatty acid compositions. The oil of *Tenebrio molitor* obtained via SFE with sc-CO_2_ presented a fatty acid profile comparable to that of black soldier fly larvae oil (Table 2), with elevated levels of the fatty acids oleic (39.80%), linoleic (33.40%), and palmitic acid (19.10%) [32]. In another study, the concentrations of these same fatty acids, for the same insect, were 39.80, 36.58, and 15.71%, respectively [30]. The observed differences were mainly attributed to the larvae cultivation process. Another insect that has a similar fatty acid profile to that of the black soldier fly is *Alphitobius diaperinus* L., whose composition is also rich in essential fatty acids [22].

The black soldier fly larvae oils extracted at 25 and 30 MPa exhibited similar concentrations of monounsaturated fatty acids (MUFAs) and varying levels of polyunsaturated fatty acids (PUFAs); unlike the SFAs, the highest levels of unsaturated fatty acids were observed at 25 MPa. The MUFA and PUFA content in black soldier fly larvae oil exceeded 50%, regardless of the process conditions. This profile is interesting because these fatty acids are known to contribute to disease prevention, especially coronary heart disease (CHD), degenerative diseases, cancer, inflammation, arthritis, and asthma [5,34]. Oleic fatty acids (omega 9, ω-9), which present antiapoptotic and anti-inflammatory properties [35,36], along with linoleic acids (omega 6, ω-6), are essential fatty acids that cannot be synthesized by mammals. These fatty acids play a role in glucose metabolism and in hypertension, as well as aid in the prevention of diabetes [37]. The fatty acid linoleic acid, also known as ALA (α-linoleic acid), presented levels of 2.21 and 2.84% for the different pressures used in the process (25 and 30 MPa, respectively) (Table 2), which were higher than those found in other insects. The daily consumption of this fatty acid increases the concentrations of eicosapentaenoic acid/docosahexaenoic acid (EPA/DHA) in human plasma [38].

The elevated concentrations of unsaturated fatty acids in the oil of *Hermetia illucens* L., resembling the characteristics of some vegetable oils, such as maize, soybean, and olive oil, suggest that insect oils can offer health benefits. Research focusing on their minor compounds can help determine how they can be incorporated into human nutrition.

The black soldier fly oils obtained in other studies showed higher concentrations of saturated fatty acids [36,39,40] coinciding with the profile of the oil extracted at 25 MPa (Table 2). However, when SFE was conducted at 30 MPa, the highest levels of fatty acids observed were unsaturated fatty acids, resulting in oils with greater potential health benefits. Therefore, it appears that elevated pressures in the extraction process can increase the levels of unsaturated fatty acids.

The triacylglycerols (TAGs) found in *Hermetia illucens* L. oil obtained via SFE with sc-CO_2_ and ethanol as a cosolvent exhibited a similar composition to the oil obtained via SFE with only sc-CO_2_ [5]. This behavior was expected because the proportion of fatty acids also remained similar. The algorithmic prediction of TAGs indicated that those found in greater concentrations all contained at least one unsaturated fatty acid in their composition (Table 3), a characteristic influenced by the fatty acid composition (Table 2). At room temperature, this oil is found in liquid form, but when refrigerated (−4 °C), it solidifies. Because it contains TAGs with unsaturated fatty acids, the composition of this oil resembles both animal and vegetable oils.

The probable triacylglycerols (TAGs) in black soldier fly larvae oil extracted via SFE with sc-CO_2_ and ethanol as a cosolvent at 60 °C and 30 MPa, observed in higher concentrations, were those composed of lauric, oleic, and palmitic (LaOP) fatty acids (11.18%), followed by TAGs composed of myristic, oleic, and palmitic (MOP) acid (8.52%), lauric, linoleic, and palmitic (LaLiP) acid (8.24%), myristic, linoleic, and palmitic (MLiP) acid (8.00%), and palmitic, linoleic, and palmitic (PLiP) acid (7.62%) (Table 3).

The TAGs obtained via SFE with sc-CO_2_ and ethanol as a cosolvent at 60 °C and 25 MPa were the same as those observed at 30 MPa; however, their concentrations varied. Higher levels were observed in TAGs composed of lauric, oleic, and palmitic (LaOP) fatty acids (9.03%), followed by myristic, linoleic, and palmitic (MLiP) acid (7.85%), palmitic, linoleic, and palmitic (PLiP) acid (7.82%), lauric, linoleic, and palmitic (LaLiP) acid (7.57%), and myristic, oleic, and palmitic (MOP) acid (7.13%). This difference was attributed to the higher levels of unsaturated and monounsaturated fatty acids observed in the extraction at 25 MPa.

The oils from *Hermetia illucens* L. larvae obtained via SFE without cosolvent exhibited different TAG concentrations for LaLiP (8.42%) and MOP (7.34%) [5]. When calculating the number of probable TAGs, those with a mass % greater than 0.5% were considered. Meanwhile, in the oil obtained via SFE using ethanol as a cosolvent, another TAG was considered, trilinolein (LiLiLi) (Table 3), which was different from the oil obtained with only sc-CO_2_.

Knowledge of TAG molecular species can provide valuable insights on lipid properties, such as the melting point range, the solid fat index, and crystalline structure. These physical properties affect the organoleptic properties of foods. In addition, they are fundamental for understanding the oxidative properties of oils [41].

The statistical method used provides information on the probable TAG content present in the oil in function of its composition in fatty acids. Nevertheless, oils with similar fatty acid compositions may not always contain the same TAGs. African palm weevil (*Rhynchophorus ferrugineus*) oils, for example, have similar fatty acid contents to black soldier fly larvae oil; however, among the 17 TAGs identified in their composition, the highest levels were of POO (36.4%) and PPO (30.6%) [42]. The analytical methods used for analyzing TAGs are more accurate than statistical ones.

#### 3.3.2. Identification of Free Fatty Acids in the Oil Obtained via Extraction with sc-CO_2_ Using Ethanol as a Cosolvent

The oil of *Hermetia illucens* L. obtained via SFE with sc-CO_2_ using ethanol as a cosolvent presented a similar free fatty acid profile to the oil obtained when only sc-CO_2_ was used as a solvent [5]. The percentage by area of free fatty acids present in the oil was greater for lauric (12:0, La), palmitic (16:0, P), linoleic (18:2n -6, Li), oleic (18:1n-9, O), and myristic acid (14:0, M) (Table 2). Free fatty acid contents are related to an oil’s acidity, which can cause lipid degradation via oxidation, resulting in alterations in the organoleptic properties of the oil [43,44].

In the present study, the *Hermetia illucens* L. oils extracted under different pressure conditions exhibited varying free fatty acid content values (Table 2). The interference of pressure, in this case, should be disregarded because this method is considerably qualitative and only enables the identification of free fatty acids in the oil, not their quantification. Normally, the free fatty acids observed in higher concentrations are also found in higher concentrations when determining the fatty acid profile.

The free fatty acid index is indicative of the quality of the oil because it measures the rate of oxidation. In other words, the lower the free fatty acid content, the better the quality of the oil, which is why controlling the acidity of an oil is crucial during processing.

#### 3.3.3. Nutritional Quality Indices of the Lipids

The evaluation of the nutritional values of the lipids in the black soldier fly larvae oils extracted via SFE with sc-CO_2_ and ethanol under different pressure conditions revealed high levels of desirable fatty acids (DFAs), as well as high relative values for the ratio between polyunsaturated and saturated fatty acids (PUFAs/SFAs) (Table 4). DFAs and the PUFAs/SFAs ratio contribute to the prevention of certain cardiovascular diseases and some types of cancer, highlighting the advantages of incorporating these oils into the human diet. The recommended PUFAs/SFAs ratio in foods is considered to be above 0.45% [45].

The PUFAs/SFAs ratio in black soldier fly larvae oil was found to be similar to that observed in chicken breast, which is considered a lean meat that is recommended for low-calorie diets [46]. However, when compared to pork meat (0.13) and coconut oil (0.26%), the values were higher (Table 4).

Hypercholesterolemic acids (OFAs) were found in lower concentrations compared to DFAs (Table 4), which is a positive health-related characteristic. The OFA values in insect oils are similar to those observed in sunflower and soybean vegetable oils [47]. This similarity is due to the high levels of oleic and linoleic acids in black soldier fly larvae oils, contributing to their healthiness.

The nutritional value indices (NVI) of the *Hermetia illucens* L. larvae oils obtained via SFE under both pressure conditions were found to be similar to those observed in *Alphitobius diaperinus* L. oils (1.24–1.88%) [22]. These values are associated with high concentrations of polyunsaturated and monounsaturated fatty acids, which are known to contribute to the prevention of several heart diseases. The ratio between the concentrations of hypocholesterolemic and hypocholesterolemic (h/HH) fatty acids in the oil of black soldier fly larvae obtained via SFE with sc-CO_2_ and ethanol as a cosolvent was 1.16 and 0.97% for the oils obtained at 25 and 30 MPa, respectively (Table 4). These values were similar to those found in pequi fruit oil, an exotic oil consumed for its bioactivity due to its composition, with h/HH ratios ranging from 1.01 to 1.04% [48].

The higher the values of the h/HH ratio, the greater the benefits for human health. This is because fatty acids with hypocholesterolemic (h) properties assist in cholesterol metabolism [21] and, consequently, in the prevention of atherosclerosis. This relationship is highly significant when assessing the nutritional quality of oils and fats.

The atherogenic indices (AI) refer to the ratio between the sum of atherogenic saturated fatty acids, which strengthen the immune and circulatory systems, and atherogenic unsaturated fatty acids of different types, which aid in reducing esterified fatty acids and preventing coronary diseases. On the other hand, the thrombogenic index (TI) indicates the propensity for the development of blood vessel clots; therefore, the lower the AI and TI values, the greater the risk of cardiovascular diseases [45].

The black soldier fly larvae oils obtained via SFE using ethanol as a cosolvent (10%) exhibited AI values of 1.25 and 1.49% and TI values of 0.73 and 0.91% for the pressure conditions of 25 and 30 MPa, respectively. Meanwhile, the oil of “Tucumã stone bug” larvae (*Speciomerus ruficornis*), an insect native to the Amazon region, presented an AI of 3.06% and a TI of 2.17% [49], which were much higher than the values observed in this study. Although the “Tucumã stone bug” larva is an insect, its fatty acid composition differs from that of black soldier fly larvae, which is why there are discrepancies in the nutritional indices. Additionally, the oil of the lesser mealworm beetle has a TI of 1.04%, a higher value than that of black soldier fly larvae oil, and a lower AI, of 0.53%.

The *Hermetia illucens* L. larvae oil obtained via SFE with sc-CO_2_ and ethanol (10%) as a cosolvent presented a PUFA ω-6/ω-3 ratio value of approximately 7% (7.1–7.9%) (Table 4), a value similar to that of wheat germ vegetable oil (7.4%) and lower than that of *Alphitobius diaperinus* L. oil, whose values range from 15.48 to 17.54% [45]. High values for this index (PUFAs ω-6/ω-3 ratio) are not recommended, with ratios below 4.0% considered ideal. Therefore, when considering this nutritional index, the black soldier fly larvae oil obtained via SFE with sc-CO_2_ and ethanol (10%) would not be recommended for consumption. The calculated oxidizability (COX) index indicates the levels of oxidizable matter in the oil or its susceptibility to oxidization. In the black soldier fly larvae oil, the COX values were 2.52 and 2.91% (Table 4), which is a similar rate to apricot seed oil (3.3%) [50]. This similarity is due to the fact that both oils contain comparable amounts of polyunsaturated fatty acids. The COX index is directly associated with the unsaturated fatty acid content, meaning that the greater the number of double bonds, the higher the susceptibility to oxidation.

The oxidative induction time, which was determined rapidly via the Rancimat method, of the sample of black soldier fly oil obtained via SFE with sc-CO_2_ + EtOH (10%) was relatively short (0.02 h) (Table 4). In the extraction via SFE with sc-CO_2_ without ethanol as the cosolvent, the same result was observed, indicating that the extraction method or the solvent used does not alter oxidative stability. Another oil that also presented low oxidative stability (0.08 h) when subjected to this test was the oil obtained from giant *Zophobas morio* via Soxhlet extraction. Lipid oxidation is a process that involves not only the free fatty acid content but also unsaturated fatty acids. The oil obtained from *Tenebrio molitor* L., extracted using pressurized n-propane, proved to be more resistant to lipid oxidation (0.91 h) when evaluated using the Rancimat method [51].

With the exception of the determination of the oil oxidation time via Rancimat, the nutritional indices were calculated based on the fatty acid content. Given that the levels of fatty acids varied slightly depending on the process conditions (25 and 30 MPa) (Table 2), variation was also expected in the values of the calculated indices (Table 4). Several other factors can alter the fatty acid content of insect oils, including species, the type of diet, sex, age, and the method of slaughter used, among others. However, because the same sample was used in different processes in this study, we attributed the differences to the specific processes, more specifically, pressure (P).

#### 3.3.4. Determination of Acidity of the Oil Obtained via SFE with sc-CO_2_ and Ethanol as a Cosolvent and of the Ethanolic Extract Obtained via PLE

Acidity levels are widely used indices that indicate the degradation of oils and are, therefore, associated with the quality of the product.

The acidity indices, expressed in mg KOH/g of oil, were considerably high in the black soldier fly larvae oil obtained via SFE with sc-CO_2_ and ethanol as a cosolvent, with values ranging from 8.01 to 9.36 mg KOH/g of oil for the pressure conditions of 25 and 30 MPa, respectively; however, they did not present a significant difference (Table 5). When only sc-CO_2_ was used as a solvent, without ethanol, the black soldier fly larvae oil exhibited an acidity index of 8.7 and 11 mg KOH/g of oil for the same pressure conditions described above [5], although with no significant differences between the extracted oils, indicating that the extraction process does not influence the acidity levels. The crude oil of *Hermetia illucens* L., which is extracted in four stages (degumming, neutralization, bleaching, and deodorization), presented a similar concentration of 11.87 mg KOH/g [38], indicating that black soldier fly oil is more acidic.

The ethanolic extract derived from the integration of the SFE (with sc-CO_2_ and without ethanol) and PLE (with pressurized ethanol) processes generated an extract with elevated acidity (48.79%) (Table 5), which was superior compared to the oils extracted via SFE with sc-CO_2_ and ethanol as a cosolvent. This result can be justified by the considerably low concentration of oil in its composition, indicating that the minor compounds present in the extract (organic acids) increase its acidity index.

#### 3.3.5. Total Phenolic Content (TPC) in the Oil Obtained via Extraction with sc-CO_2_ and Ethanol as a Cosolvent and of the Ethanolic Extract Obtained via PLE

Phenolic compounds have been shown to possess antioxidant activity. In the oil itself, they may function by inhibiting the free radicals responsible for lipid oxidation.

In the extractions of black soldier fly larvae oil via SFE with sc-CO_2_ and ethanol as a cosolvent, the total phenolic compound levels were low and were not statistically different (Table 5). The oil from the same sample of black soldier fly larvae obtained at 60 °C and 30 MPa via SFE with sc-CO_2_ without cosolvent showed even lower levels of phenolic content (0.1 ± 0.0 mg GAE/g), indicating that ethanol, which is a polar solvent, helps in the extraction of minor and functional compounds present in lipid-rich matrices. However, the phenolic content in the ethanolic extract obtained via PLE using the SFE-defatted meal with sc-CO_2_ was higher (4.7 ± 0.3) (Table 5).

When comparing the processes, the intensification of SFE with ethanol (10%) as a cosolvent helps to obtain oils rich in phenolic compounds because ethanol acts as a polarity modifier, altering the solubilization power of sc-CO_2_. However, the integration of the two processes (first the lipid extraction via SFE with sc-CO_2_, followed by PLE with pressurized ethanol) enabled the generation of another extract with different characteristics and rich in other compounds.

The oil from lesser mealworm larvae presented 4.3 mg GAE/g when extracted using ethanol/isopropanol as solvent [31]. This level of phenolic content was higher when compared to the extractions via SFE with sc-CO_2_ using ethanol (10%) alone as a solvent, and similar to the extracts obtained via SFE with sc-CO_2_ followed by PLE with pressurized ethanol, indicating that ethanol can help extract larger amounts of minor compounds.

One study showed that olive oils enriched with *Acheta domesticus* and *Tenebrio molitor* exhibited a 3.8-fold increase in phenolic content levels for the *Acheta domesticus* and a 1.7-fold increase for *Tenebrio molitor* [12].

#### 3.3.6. Total Carotenoid Content in the Oil Obtained via Extraction with sc-CO_2_ and Ethanol as a Cosolvent and of the Ethanolic Extract Obtained via PLE

Carotenoids are compounds that present antioxidant activity and are precursors of vitamin A, playing an important role in our health. The total carotenoid content in the *Hermetia illucens* L. larvae oils extracted with sc-CO_2_ using ethanol as a cosolvent at 60 °C and 25 and 30 MPa presented values of 8.2 and 8.9 mg/100 g of oil, showing no significant differences in function of pressure (Table 5). When the oil was extracted via SFE with only sc-CO_2_, the carotenoid levels were lower, ranging from 2.05 to 4.59 mg/100 g [5]. This evidences that the use of ethanol intensifies the extraction of minor compounds from oil via SFE; however, the ethanolic extract obtained from the SFE-defatted meal exhibited higher carotenoid content (11.5 mg/100 g), indicating that the integration of the two processes is more efficient in separating these compounds.

Insect oils from *Rhynchophorus ferrugineus* and *Alphitobius diaperinus* L., for example, present much lower total carotenoid contents than black soldier fly larvae, ranging between 0.6 and 0.9 mg/100 g [22,42]. Although the African palm weevil is a pigmented insect, it exhibits low levels of carotenoids, which is why it is believed that the content of these compounds in the extracts of these insects may be related to the type of diet they receive. The black soldier fly larvae used in this study were fed with restaurant waste that probably contained carotenoid-rich foods.

Quails fed with whole *Hermetia Illucens* L. meal showed an increase in color and carotenoid content in the egg yolk. This result was attributed to the elevated carotenoid content found in black soldier fly larvae meal [52].

#### 3.3.7. Tocopherol and Phospholipid Content in the Oil Obtained via SFE with sc-CO_2_ and Ethanol as a Cosolvent and of the Ethanolic Extract Obtained via PLE

Tocopherols act as antioxidants, with the function of preserving polyunsaturated lipids and other parts of the cell membrane, and are also one of the essential vitamins for our body. Vitamin E or tocopherol deficiency can lead to anemia and severe problems in newborns [53].

The total tocopherol content in the black soldier fly larvae oil obtained via SFE (sc-CO_2_+EtOH) showed different values depending on the pressure conditions used in the process, as follows: 65.45 and 68.87 mg/kg at 25 and 30 MPa, respectively (Table 5). At the higher pressure, the levels of tocopherols in the oil, as well as phenolic compounds and carotenoids, were higher, although without significant differences.

When only sc-CO_2_ was used in the SFE (60 °C and 30 MPa), the total tocopherol content in the black soldier fly larvae oil was higher (73.77 mg/kg). For these compounds, the use of a polarity modifier such as ethanol in SFEs does not affect the extraction process. The extract obtained from the defatted black soldier fly larvae meal via SFE with sc-CO_2_ showed very low tocopherol content (0.94 mg/kg), particularly α-tocopherol (Table 5). Vitamin E is fat-soluble and, therefore, is extracted with lipid compounds; thus, the ethanolic extract contains very small amounts of these compounds.

It can be noted that for all oils extracted from the black soldier fly larvae meal via SFE, with or without ethanol as a solvent, the α-tocopherol content was higher than the other tocopherols, followed by β, δ, and γ-tocopherol (Table 5).

The α-tocopherol content in the black soldier fly larvae oil obtained via SFE with sc-CO_2_ and without ethanol as a solvent, was 49.58 mg/kg, a higher value than those observed in oils obtained when ethanol (10%) was used in the extraction. α-tocopherol is the most important tocopherol, being sold by the pharmaceutical industry as a food supplement. The total tocopherol content in Alphitobius diaperinus L. oils extracted using different solvents showed higher tocopherol levels than *Hermetia illucens* L. oils, mainly γ-tocopherol [22]. Oil from the *Rhynchophorus ferrugineus*, extracted using a mixture of solvents (chloroform: methanol: distilled water), showed higher total tocopherol content [44], indicating that this tocopherol extraction method using combined solvents increases the concentrations of tocopherols in insect oils.

The black soldier fly larvae oils extracted via SFE (sc-CO_2_ + EtOH) exhibited phospholipid contents of 2.4 and 3.0 mg/100 g, with slightly higher levels in the oil obtained when the pressure of the process was 30 MPa. The greater extraction at 30 MPa may have occurred due to the increase in sc-CO_2_ solubility as a result of the increase in solvent density (78.73 kg/m^3^ at 25 MPa and 830.60 g/m3 at 30 MPa) (Table 5). The influence of ethanol is notable, as the change in solvent polarity promotes greater extraction of these compounds, with a yield of 1.5 mg/100 g when using only sc-CO_2_ (60 °C and 30 MPa) as solvent.

The ethanolic extract of the black soldier larvae meal, which was defatted using SFE (sc-CO_2_) and obtained via PLE, also presented a concentration of 1.5 mg/100 g of phospholipids (Table 5), which, added to the 1.5 mg/100 g extracted with sc-CO_2_ at 60 °C and 30 MPa, results in 3.0 mg/100 g extracted via SFE using ethanol as a cosolvent. Regarding the phospholipid content, in particular, the intensification of the SFE process by the addition of 10% ethanol as a polarity modifier assisted in their extraction.

According to one study, different insect oils extracted using organic solvents (dichloromethane; methanol; petroleum ether) and thin-layer chromatography evidenced the presence of phospholipid content; however, the study did not quantify these concentrations. When the authors used water as a solvent, the obtained extracts did not present phospholipids in their composition [6]. Ochiai & Komiya [54] determined the phospholipid content in different insects (powdered crickets [*Acheta domestica*, *Gryllus assimilis*, and *Gryllus bimaculatus*], powdered migratory locust [*Locusta migratoria*], and powdered silkworm [*Bombix mori*]), which varied from 17.0 to 35.2, which were higher values than those observed in the black soldier fly larvae oil quantified in this study.

Phospholipids are critical components of the plasma membrane of plant and animal cells. Because insect oils, such as *Gryllus bimaculatus* and *Acheta domestica* oil, contain elevated levels of these compounds, they can be considered an alternative source of phospholipids [54], indicating that insect oil should be incorporated into the human diet due to its high nutritional value.

#### 3.3.8. Preliminary Determination of the Class of Compounds Present in the Ethanolic Extract of Defatted Black Soldier Fly Larvae Meal Obtained via PLE

HPTLC analyses indicated families of compounds that constitute the hydroalcoholic extract of defatted soldier fly larvae flour. It was possible to previously identify the presence of amino acids by revealing the Ninhydrin reagent (Figure 2A) and the presence of sugars, after revealing them with the Molish reagent, (Figure 2B). The standard sugar solution used was a mixture of glucose (Rf ≈ 0.51), maltose (Rf ≈ 0.40), maltopentaose (Rf ≈ 0.17), and maltoheptaose (Rf ≈ 0.08). The highest concentration of sugar present in the ethanolic extract of soldier fly larvae flour has an Rf ≈ 0.35, which is a value lower than a disaccharide (maltose) and higher than a pentose, probably a sugar with three or four monosaccharides (trioses or tetroses).

The identification of compounds that respond to the wavelength of 366 nm, which may be phenolic compounds (Figure 2C), showed distinct bands with different intensities. Phenolic compounds and carotenoid contents were quantified using colorimetric methods (Table 5). In addition, in the HPTLC analysis to identify condensed tannins, it was not possible to detect them.

High-performance liquid chromatography (HPLC) analysis indicated the presence of compounds with peaks, retention times, and maximum absorption spectra similar to those of organic acids. A detection, performed with a DAD at 280 nm indicated, in the ethanolic extract obtained by PLE of defatted black soldier fly larva flour, some peaks with maximum absorptions of 215.45, 216, 220.34, 225 and 225.22 nm (Figure 3), and the same HPLC measurement made with gallic and benzoic acids standards shows maximum absorption spectra of 217 and 221.8 nm, respectively (Figure 4).

Although we are working with preliminary analyses, the presence of organic acids in these extracts can be confirmed by their high acidity (Table 5).

### 3.4. Antioxidant Activity of the Oil Obtained via Extraction with sc-CO_2_ Using Ethanol as a Cosolvent and the Ethanolic Extract Obtained via PLE

The higher the pressure in the SFE (sc-CO_2_ + EtOH), the higher the antioxidant activity of the extract, both for the method that used DPPH and for the one that used (ABTS•+) (Table 6).

In the extracts obtained from the integration of the SFE process followed by PLE, it was expected that the ethanolic extract of the defatted black soldier fly meal (Table 6) would present greater antioxidant activity because it has higher levels of phenolic content (Figure 2C) and carotenoids in its composition (Table 5); this behavior was observed.

The antioxidant activity by DPPH of the oils extracted via SFE with sc-CO_2_+EtOH presented higher values than those extracted from *Polyrhachis vicina* ant meal [55]. The oils extracted from the *Acheta domesticus* and from *Tenebrio molitor* via PLE with ethanol also exhibited antioxidant activity, which was 80% higher when extracted via ultrasound [12]. Meanwhile, the oils obtained from *Alphitobius diaperinus* L., extracted via ultrasound with ethanol/isopropanol, presented an antioxidant activity value of 70.79% [29], while *Hermetia illucens* oil extracted via SFE with sc-CO_2_ exhibited lower values than those found herein, indicating that some factors, such as the type of solvent used and the extraction temperature and pressure, can alter the levels of extracted antioxidants.

The antioxidant activity measured using the reduction of the radical (ABTS•+) to ABTS in the *Hermetia illucens* L. oils obtained via SFE with sc-CO_2_ + EtOH presented values of 0.14 and 0.21 mMol TE/100 g of oil for the pressures of 25 and 30 MPa, respectively. The ethanolic extract obtained via PLE of the SFE-defatted black soldier fly larvae meal with only sc-CO_2_ as solvent exhibited a higher and significantly different value compared to the extracts obtained via SFE (0.72 mMol TE/100 g of oil) (Table 6). The ethanolic extract obtained from the defatted flour of black soldier fly larvae presented a value almost twice as high (0.40 mMol TE/100 g of juice) as that of orange juice [56].

The antioxidant activity measured via ABTS in *Hermetia illucens* oils extracted via SFE with sc-CO_2_ exhibited values ranging from 0.03 to 0.05 mMol TE/100 g, which were lower than those observed in this study [40].

This discrepancy can be attributed to the low pressures and temperatures used in the extraction, which was conducted without the use of a cosolvent, suggesting that these factors can alter the concentrations of antioxidants in insect oils. The black soldier fly larvae oils presented high levels of substances capable of inhibiting oxidation; therefore, further research on the use of the ethanolic extract as an antioxidant product is warranted. Natural products have garnered significant interest from food processing and cosmetics companies, as they seek natural antioxidants to replace synthetic ones because natural antioxidants can reduce risks to consumers’ health.

## 4. Conclusions

The extractions with sc-CO_2_ using ethanol as a solvent (sc-CO_2_ + EtOH) (10%) displayed variations in the yields obtained under different pressure conditions, with higher values being observed when the pressure was set at 30 MPa while maintaining the same temperature (60 °C). The extraction with cosolvent (EtOH) exhibited higher efficiency compared to the SFE using sc-CO_2_ alone, suggesting that altering the solvent’s polarity led to the extraction of additional compounds.

The SFE using sc-CO_2_ and ethanol as a polarity modifier is also quite efficient at defatting black soldier fly larvae meal. The advantage of intensifying the SFE process using sc-CO_2_ + EtOH was that the extracted oil presented higher levels of minor compounds, which resulted in higher concentrations of phenolic compounds and carotenoids and greater antioxidant activity. In this instance, the intensification of the process proved to be quite valid.

The integration of supercritical fluid extraction (SFE) with subcritical carbon dioxide (sc-CO_2_), followed by pressurized liquid extraction (PLE) using ethanol yields two distinct extracts, including an oil and an ethanolic extract. The ethanolic extract is notably richer in phenolic compounds, carotenoids, and antioxidant activity compared to the oil obtained via sc-CO_2_ + EtOH. Consequently, defatted black soldier fly larvae meal purified with pressurized ethanol emerges as a product abundant in proteins and fibers, currently under study for food protein enrichment. In this process integration, sc-CO_2_ solubilizes lipid compounds and lipid-free meal, while pressurized ethanol extraction allows for the solvation of polar compounds remaining in the meal, particularly phenolic compounds (4.7 mg GAE/g of extract) and carotenoids (11.5 mg/100 g of extract).

Ethanol also functions similarly in supercritical fluid extraction (SFE), where the solvents (sc-CO_2_ + EtOH) dissolve all compound classes in black soldier fly larvae meal. However, in process integration, ethanol selectively extracts compounds left in the matrix, excluding lipids. Intensifying SFE with ethanol or integrating SFE with PLE produces products suitable for cosmetics (oil) and food production (oil and defatted protein meal) industries. These processes, free from toxic solvents or unsustainable compounds, yield oils enriched with minor compounds and defatted meal with high protein, fiber, and amino acid content, devoid of toxic residues.

The crude oil of black soldier fly larvae oil obtained via sc-CO_2_ + EtOH at 25 and 30 MPa was also characterized regarding its acidity, which was found to high, with lauric acid being the predominant free fatty acid.

The fatty acid composition of oils from SFE with sc-CO_2_ + EtOH varied slightly with different pressures. Oils at 25 MPa had lower saturated fatty acid (SFA) content (46.48%) compared to those at 30 MPa (50.46%). Even minor process variations can affect fatty acid content, influencing nutritional quality indices. Black soldier fly oil showed favorable health properties with elevated desirable fatty acids (DFAs) and a high polyunsaturated to saturated fatty acid (PUFA/SFA) ratio. These factors are linked to preventing cardiovascular diseases and cancer. The recommended PUFA/SFA ratio (>0.45%) was observed in oils at 25 MPa.

The probable TAGs extracted with sc-CO_2_ + EtOH exhibited higher levels in the oil containing LaOP and MOP fatty acids, both with values of around 19.70%, which reflects the greater levels of fatty acids in their composition. This oil also presented high levels of phospholipids and tocopherols.

This study showed that the supercritical fluid extraction (SFE) process, or its conditions, can alter the fatty acid composition and the presence of minor bioactive compounds in the extracts obtained. Additionally, integrating SFE with pressurized liquid extraction (PLE) yielded an ethanolic extract with higher acidity, elevated carotenoid and total phenolic compound levels, and the presence of amino acids, sugars, phenolic compounds, and organic acids.

## Figures and Tables

**Figure 1 foods-13-01620-f001:**
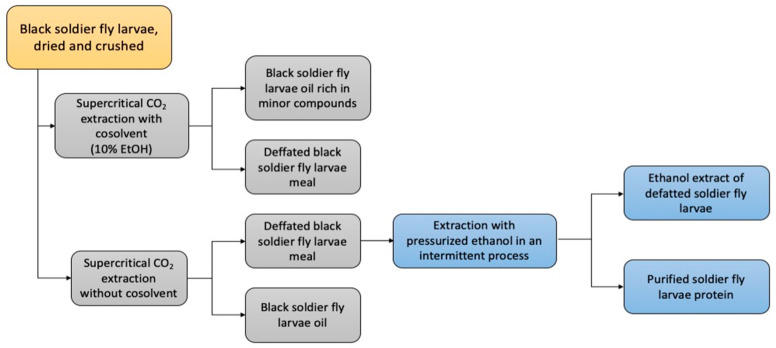
Integration of green extraction processes for obtaining purified extracts of black soldier fly larvae meal.

**Figure 2 foods-13-01620-f002:**
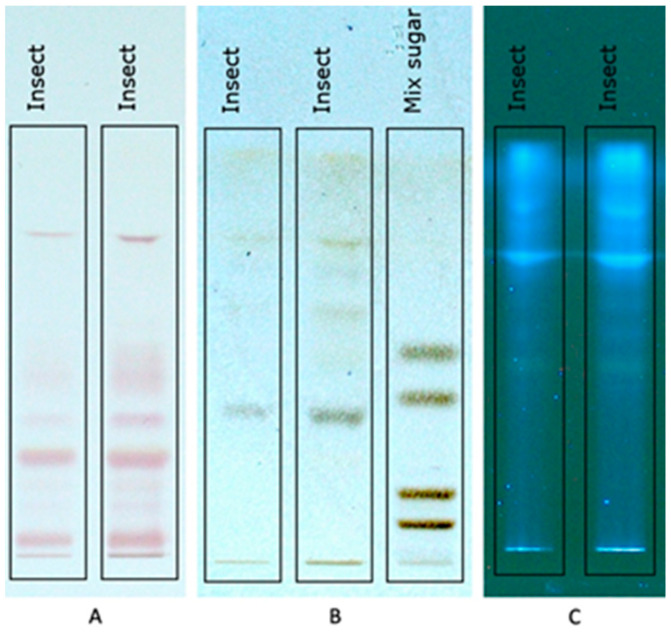
HPTLC plates; (**A**) indicates the presence of amino acids with Ninhydrin as a developer; (**B**) indicates the presence of sugars with development using Molish reagent; (**C**) indicates the presence of compounds with fluorescence at 366 nm with mobile phase ethanol/ammonium hydroxide/water (75.5:12.5:12).

**Figure 3 foods-13-01620-f003:**
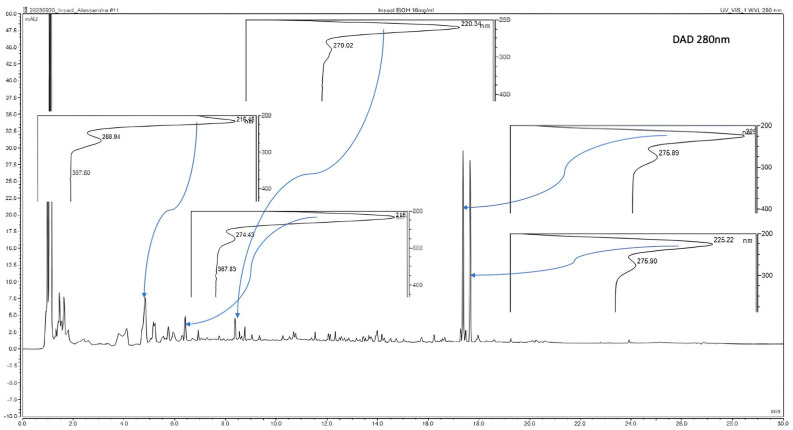
Chromatogram of the ethanolic extract of defatted flour from black soldier fly larvae with some maximum absorption spectra (UHPLC, Thermo Scientific Ultimate 3000 RSLC analytic system) reading at 280 nm.

**Figure 4 foods-13-01620-f004:**
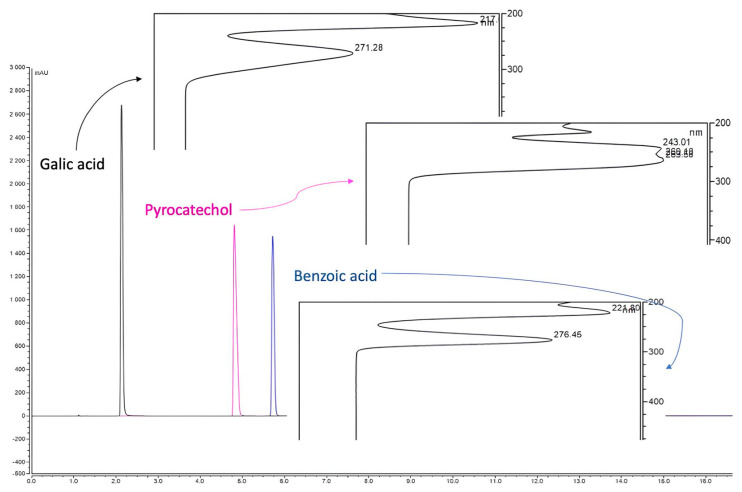
Chromatogram of standard compounds (galic and benzoic acids, purocatechol) with maximum absorption spectra (UHPLC, Thermo Scientific Ultimate 3000 RSLC analytic system) reading at 280 nm.

**Table 1 foods-13-01620-t001:** Yields of the SFE using ethanol as a cosolvent (10%) (sc-CO_2_+EtOH) and the extract obtained via PLE with pressurized ethanol from defatted meal (by SFE*, 60 °C and 30 MPa [5]).

SFE** (sc-CO_2_ + EtOH)			
Yield (%)		Oil Recovery ***(%)	
32.5 ^b^ ± 0.1 (25 MPa, 60 °C)		95.8 ± 0.3 ^b^	
36.1 ^a^ ± 0.2 (30 MPa, 60 °C)		106.5 ± 0.6 ^a^	
PLE (EtOH)			
Yield PLE (%)	Total Yield (%) (Yeld by SFE*, 60 °C and 30 MPa [5] + Yield PLE)		Oil Recovery *** (%) (SFE* + PLE)
20.9 ± 0.5	(33 + 20.9) 53.9		157

SFE*: yield of the oil extracted using sc-CO_2_ without cosolvent [5], which generated the defatted meal used in the PLE process to obtain the ethanolic extract. SFE**: extraction conditions (sc-CO_2_ + EtOH) 1.5 h at constant flow of 10 g/min of CO_2_, S/F of 90 g CO_2_/g of black soldier fly meal. Oil Recovery ***: it is the recovery of the extract compared to the total lipid content present in the sample (32.7%). Different letters in the same column show a significant difference (*p* < 0.05).

**Table 2 foods-13-01620-t002:** Fatty acid profile and free fatty acid profiles of black soldier fly larvae meal oil obtained via SFE using ethanol as a cosolvent at 60 °C and 25 and 30 MPa.

Fatty acid profile				
Pressures (MPa)			25	30
Peak	RT (min)	Fatty Acids	Area %	Area %
1	15.81	10:0 (Capric) (C)	0.59 ± 0.05	0.66 ± 0.03
2	19.10	12:0 (Lauric) (La)	11.35 ± 0.32	13.00 ± 1.76
3	22.31	14:0 (Myristic) (M)	8.63 ± 0.43	9.61 ± 0.14
4	25.65	16:0 (Palmitic) (P)	21.09 ± 0.44	22.31 ± 0.16
5	29.07	18:0 (Stearic) (S)	4.81 ± 0.07	4.92 ± 0.17
		SFA	46.48 ± 1.24 ^a^	46.48 ± 1.24 ^a^
6	26.60	16:1 (Palmitoleic) (Po)	6.11 ± 0.13	5.91 ± 0.22
7	29.76	18:1 ω-9 (Oleic) (O)	24.64 ± 0.56	24.00 ± 0.31
8	31.60	18:2 ω-6 (Linoleic) (Li)	19.94 ± 0.71	17.54 ± 0.83
9	33.52	18:3 ω-3 (Linolenic) (Ln)	2.84 ± 0.10	2.21 ± 0.19
		MUFA	30.74 ± 0.43 ^a^	29.79 ± 0.64 ^a^
		PUFA	22.78 ± 1.24 ^a^	19.74 ± 1.02 ^a^
Free Fatty Acids				
2	19.56	12:0 (La)	2.23 ± 0.31	5.86 ± 0.49
3	22.94	14:0 (M)	0.35 ± 0.07	0.45 ± 0.07
4	26.47	16:0 (P)	1.52 ± 0.25	4.17 ± 0.08
7	30.92	18:1 ω-9 (O)	1.31 ± 0.27	2.87 ± 0.21
8	32.31	18:2 ω-6 (Li)	1.24 ± 0.01	1.59 ± 0.02

Legend: RT = retention time, SFA = saturated fatty acids; MUFA = monounsaturated fatty acids; PUFA = polyunsaturated fatty acids. Different superscript letters in the same row indicate a significant difference (*p* < 0.05).

**Table 3 foods-13-01620-t003:** Probable triacylglycerol composition of black soldier fly larvae oil obtained via SFE using sc-CO_2_ at 30 MPa and 60 °C and ethanol (10%) as a cosolvent.

Group (X:Y) ^a^	Triacylglycerol	Molar Mass (g/mol)	Molar (%)	Mass (%)
C40:1	LaPoLa	693.10	1.06	1.23
C42:1	LaOLa	721.16	3.94	4.38
C42:2	LaLiLa	719.14	2.13	2.37
**C44:1**	**LaOM**	**749.21**	**6.44**	**6.89**
C44:2	LaLiM	747.19	3.19	3.42
**C46:1**	**LaOP**	**777.26**	**10.84**	**11.18**
**C46:2**	**LaLiP**	**775.25**	**7.97**	**8.24**
C46:3	LaLnP	773.23	1.82	1.89
**C48:1**	**MOP**	**805.32**	**8.56**	**8.52**
**C48:2**	**MLiP**	**803.30**	**8.01**	**8.00**
C48:3	LaLiO + LaOLi	801.28	4.22	4.22
C48:4	LiLiLa	799.27	1.57	1.58
**C50:1**	**POP**	**833.37**	**7.25**	**6.97**
**C50:2**	**PLiP**	**831.35**	**7.90**	**7.62**
C50:3	MOLi + MLiO	829.34	3.91	3.77
C50:4	LiLiM	827.32	1.22	1.18
C52:1	POS	861.42	2.35	2.19
C52:2	POO	859.41	4.97	4.64
C52:3	POLi + PLiO	857.39	5.13	4.80
C52:4	LiLiP	855.38	2.65	2.49
C52:5	PLiLn + PLnLi	853.36	0.64	0.60
C54:2	OOS	887.46	0.78	0.71
C54:3	SOLi + SLiO	885.45	1.31	1.19
C54:4	OOLi	883.43	1.31	1.19
C54:5	LiLiO	881.41	0.83	0.76

^a^ X represents the number of carbons (excluding glycerol carbons), and Y is the number of double bonds La = lauric acid; M = myristic acid; P = palmitic acid; Po = palmitoleic acid; S = stearic acid; O = oleic acid; Li = linoleic acid; Ln = linolenic acid.

**Table 4 foods-13-01620-t004:** Nutritional indices of the black soldier fly larvae oils obtained via SFE with sc-CO_2_ and ethanol (10%) as a cosolvent, calculated based on the fatty acid profile.

Nutritional Indices	Content * (%)	
Desirable Fatty Acids (DFAs)	58.34 ± 1.24 ^a^	54.46 ± 1.76 ^b^
PUFAs/SFAs	0.49 ± 0.03 ^a^	0.39 ± 0.03 ^b^
Hypercholesterolemic Fatty Acids (OFAs)	41.08 ± 1.14 ^a^	44.88 ± 1.76 ^b^
Nutritional Value Index (NVI)	1.40 ± 1.24 ^a^	1.29 ± 0.03 ^b^
Hipocholesterolemics/Hypercholesterolemics (h/HH)	1.16 ± 0.07 ^a^	0.97 ± 0.07 ^b^
Atherogenic Index (AI)	1.25 ± 0.07 ^a^	1.49 ± 0.09 ^b^
Thrombogenic Index (TI)	0.73 ±0.04 ^a^	0.91 ± 0.09 ^b^
PUFAs ω-6/ω-3	7.01 ± 0.05 ^a^	7.96 ± 0.33 ^b^
COX	2.91 ± 0.10 ^a^	2.52 ± 0.13 ^b^
Oxidative Stability (h)	0.02 ± 0.02	

* Results with the same superscript letters in a same row do not differ from each other according to Tukey’s mean comparison test at 5% of significance.

**Table 5 foods-13-01620-t005:** Acidity index, total phenolic content, total carotenoids, tocopherols, and phospholipids of the black soldier fly larvae oils obtained via SFE with sc-CO_2_ and ethanol (10%) as a cosolvent and of the extract obtained from the SFE-defatted larvae meal via PLE with pressurized ethanol.

	SFE (sc-CO_2_ + EtOH) 60 °C		PLE
	25 MPa	30 MPa	
Acidity index (mg KOH/g)	8.01 ± 0.04 ^a^	9.36 ± 0.63 ^a^	48.79 ± 0.84 ^b^
Total Phenolic Content (mg GAE/g)	0.3 ± 0.0 ^b^	0.5 ± 0.0 ^b^	4.7 ± 0.3 ^a^
Total carotenoids (mg/100 g)	8.2 ± 0.0 ^b^	8.9 ± 0.5 ^b^	11.5 ± 0.9 ^a^
Total tocopherols (mg/kg)	65.45	68.87	0.94
α-Tocopherol (mg/kg)	36.18	40.47	0.94
β-Tocopherol (mg/kg)	10.68	11.14	<0.50 (LQ)
γ-Tocopherol (mg/kg)	1.72	1.58	<0.50 (LQ)
δ-Tocopherol (mg/kg)	16.87	15.68	<0.50 (LQ)
Phospholipids (mg/100 g)	2.4 ± 0.0	3.0 ± 0.0	1.50

Same superscript letters in a same row do not differ from each other according to Tukey’s mean comparison test at 5% significance.

**Table 6 foods-13-01620-t006:** Antioxidant activity of the black soldier fly larvae oil obtained via SFE at 60 °C using sc-CO_2_ without and with ethanol (10%) as a cosolvent (sc-CO_2_ + EtOH) and of the ethanolic extract of the sc-CO_2_-defatted meal obtained via PLE.

Method	SFE			PLE
	sc-CO_2_	sc-CO_2_ + EtOH		
	30 MPa	25 MPa	30 MPa	
DPPH (%)	58.90 ± 2.71 ^c^	74.05 ± 2.18 ^b^	83.21 ± 0.93 ^a^	84.21 ± 1.50 ^a^
ABTS (mMol TE/100 g)	0.05 ± 0.00 ^d^	0.14 ± 0.01 ^c^	0.21 ± 0.00 ^b^	0.72 ± 0.01 ^a^

Same superscript letters in the same row do not differ from each other according to Tukey’s mean comparison test at 5% significance for the same oil.

## Data Availability

The original contributions presented in the study are included in the article, further inquiries can be directed to the corresponding author.

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
