# Peer review of "Intensification of the SFE Using Ethanol as a Cosolvent and Integration of the SFE Process with sc-CO2 Followed by PLE Using Pressurized Ethanol of Black Soldier Fly (Hermetia illucens L.) Larvae Meal—Extract Yields and Characterization"

_foods, 2024, doi:10.3390/foods13111620_

Round 1

Reviewer 1 Report

Comments and Suggestions for Authors

The black soldier fly is a valuable resource insect whose larvae have been shown to be effective manure recyclers. In addition to being a good source of oil and protein for animal feed, black soldier fly larvae have the potential to convert organic waste into a rich fertiliser. The paper is well written and presented. However, it cannot be accepted in its current form.

Some issues:

Revise the full text and reduce the repetition rate, especially for this paper (Oil extraction from black soldier fly (Hermetia illucens L.) larvae meal by dynamic and intermittent processes of supercritical CO2-Global yield, oil characterization, and solvent consumption, https://doi.org/10.1016/j.supflu.2023.105861)

line 32-33 please provide the citation.

line 38-41 please provide the citation.

line 48 Species should be written with their scientific name, when they first appear in the text. not in line 99.

line 55-57 please provide the citation.

line 62 CO2 It's supposed to be a footnote, and should be consistent throughout the text.

line 136 The equation 1 should be revised.

line 145, 184,186, 187 etc. Please check the full text for superscripts, footers, etc.

line 169 Figure 1 should be changed to better fit the layout.

line 275 Please explain 56.1 in equation 9

line 306 The equation 10 should be re-edited

line 860, 863, Figure 3 and Figure 4 are obtained by UHPLC, please added apparatus in figure captions.

The figures of GC-MS are lack in this manuiscript, please added them, and compared with the tables.

line 904 Conclusions is too long. Please summarise the ideas in the manuscript.

line 904 Lack of a discussion section.

Also, please add information as appropriate.

Author Contributions, Institutional Review Board Statement, Informed Consent Statement, Data Availability Statement, Conflicts of Interest

Author Response

Por favor, verifique o anexo

Reviewer 2 Report

Comments and Suggestions for Authors

The comments are as follows:

1. Overall conclusion is missing in the Abstract section.

2. More information about the raw material is required in the section Materials and methods. Moisture content of the dried sample material should be added. 

3. The authors stated that the they used optimised extraction conditions for PLE. The optimisation process is missing and should be added or justified.

4. Why was the defeated black soldier fly larvae meal not used to evaluate the content of polar compounds in it after applying SFE with ethanol? 

5. Why did the authors choose DPPh and ABTS assays for antioxidant activity determination?

6. Why was the characterization of the protein fraction not done in order to prove?

7. Table 1 is quite confusing. What means total extract recovery and what is its relevance? How the oil recovery was calculated, especially with its value higher that 100%? Please, make the table more understandable.

8. Line 419-436. Please check the values for yields if they are correct. The relevance of this part has to be more justified. What is the relevance of the total extract recovery?

9. Why was the identification and quantification of the polar fraction not done?

10. The nine self-citations of the co-author Alessandra Lopes de Oliveira should be revised and reduced.

11. Please, check the text for typos and grammatical errors.

Round 2

Reviewer 2 Report

Comments and Suggestions for Authors

The comments:

1. "This demonstrates that the SFE process, or even its conditions (within the same process), can alter the composition of fatty acids and minor bioactive compounds in the extracts obtained." This sentence is not clear. Is it a conclusion of the whole study?

2. Subsection 2.1.: How is the raw material provided? What is his origin? How it is treated before purification?

3. What is the relevance of the total extract recovery?

4. The authors should remove the self citations of references 14 and 16. For already well-known facts, it is not necessary to cite the literature, especially for each sentence different sources. Also, cited references should be relevant to the highlighted text.
